# Linking medicinal cannabis to autotaxin–lysophosphatidic acid signaling

Mathias C Eymery[1] , Andrew A McCarthy[1] , Jens Hausmann[1,2]

**Autotaxin is primarily known for the formation of lysophosphatidic acid (LPA) from lysophosphatidylcholine. LPA is an important signaling phospholipid that can bind to six G protein–coupled receptors (LPA$_{1-6}$). The ATX-LPA signaling axis is a critical component in many physiological and pathophysiological conditions. Here, we describe a potent inhibition of $\Delta^9$-*trans*-tetrahydrocannabinol (THC), the main psychoactive compound of medicinal cannabis and related cannabinoids, on the catalysis of two isoforms of ATX with nanomolar apparent EC$_{50}$ values. Furthermore, we decipher the binding interface of ATX to THC, and its derivative 9(R)-Δ6a,10a-THC (6a10aTHC), by X-ray crystallography. Cellular experiments confirm this inhibitory effect, revealing a significant reduction of internalized LPA$_1$ in the presence of THC with simultaneous ATX and lysophosphatidylcholine stimulation. Our results establish a functional interaction of THC with autotaxin–LPA signaling and highlight novel aspects of medicinal cannabis therapy.**

## Introduction

Autotaxin (ATX; ENPP2) is an extracellular glycoprotein, which hydrolyzes lysophosphatidylcholine (LPC) into lysophosphatidic acid (LPA) by cleaving off the choline head group (Tokumura et al, 2002; Umezu-Goto et al, 2002; Moolenaar & Perrakis, 2011). LPA is a multifunctional bioactive lipid mediator with six designated G protein–coupled receptors (LPA$_{1-6}$) (Noguchi et al, 2009), forming together with ATX the ATX-LPA signaling axis. ATX is the main producer of LPA in blood, which has been demonstrated by a heterozygous *Enpp2* (ATX) knockout mouse model. These mice show only 50% of the normal LPA levels in serum (van Meeteren et al, 2006). It is widely accepted that the ATX-LPA signaling axis is of crucial importance for lipid homeostasis in humans (Smyth et al, 2014). ATX is present in almost every body fluid and essential for murine embryonic vessel formation, which highlights its importance for life (Aoki, 2004; Boutin & Ferry, 2009; Moolenaar & Perrakis,

2011). Thus, the ATX-LPA axis is linked to numerous physiological and pathological processes, such as vascular and neuronal development, neuropathic pain, fibrosis, and immune-mediated diseases including rheumatoid arthritis, multiple sclerosis, atherosclerosis, and cancer (Moolenaar & Perrakis, 2011). In fact, *Enpp2* (ATX) is among the top 40 most up-regulated genes in metastatic breast cancer (Euer et al, 2002), whereas ATX-LPA signaling is positively correlated with the invasive and metastatic potential of several cancers including melanoma, breast, ovarian, thyroid, renal cell, lung, neuroblastoma, hepatocellular carcinoma, and glioblastoma multiforme (Samadi et al, 2011).

ATX consists of four domains, two repetitive N-terminal somatomedin B–like domains (SMB1 and SMB2), followed by the catalytic phosphodiesterase domain (PDE) and an inactive nuclease domain (Nuc) (Hausmann et al, 2011; Nishimasu et al, 2011). The active site of ATX constitutes a bimetallo zinc coordination center and the active site nucleophile, Thr209, in rodents (Hausmann et al, 2011). A nearby hydrophobic pocket, which extends into the PDE domain, accommodates the lipid substrate aliphatic chain; in addition, there is an allosteric tunnel that is formed between the SMB2 and PDE domains, where an oxysterol and bile acids bind (Hausmann et al, 2011; Keune et al, 2016).

The gene product of ATX can exist in at least three different isoforms, which are ATX-$\alpha$, ATX-$\beta$, and ATX-$\gamma$, as a result of an alternative splicing event (Giganti et al, 2008). ATX-$\alpha$ is characterized by a polybasic insertion of 52 amino acids in the PDE domain, when compared to the canonical plasma isoform ATX-$\beta$. ATX-$\alpha$ can bind to heparin and cell surface heparan sulfate proteoglycans (Houben et al, 2013), whereas ATX-$\beta$ has been shown to bind to $\beta_1$ and $\beta_3$ subunits of integrins (Kanda et al, 2008; Hausmann et al, 2011). ATX-$\gamma$ is the so-called "brain-specific" isoform (Perrakis & Moolenaar, 2014) and has been implicated with neuronal disorders, such as multiple sclerosis, depression, Alzheimer's disease, and neuropathic pain (Moolenaar & Perrakis, 2011).

Another important signaling system is the well-established endocannabinoid system (Cristino et al, 2020), with its two cannabinoid receptors, the cannabinoid receptor type 1 (CB$_1$) and type 2 (CB$_2$) (Matsuda et al, 1990; Munro et al, 1993). The human CB$_1$ is primarily expressed in the central nervous system and also present in the peripheral nervous system and testis (Matsuda et al, 1990), whereas the CB$_2$ is mainly expressed in the immune system (Munro

---

[1]European Molecular Biology Laboratory, Grenoble, Grenoble, France    [2]European Molecular Biology Laboratory, Chemical Biology Core Facility, Heidelberg, Germany

Correspondence: andrewmc@embl.fr; jens.hausmann@uni-oldenburg.de
Jens Hausmann's present address is Research Group Anatomy, School of Medicine and Health Sciences, Carl von Ossietzky University of Oldenburg, Oldenburg, Germany

 

et al, 1993). The endogenous ligands, anandamide (Devane et al, 1992) and 2-arachidonoylglycerol (2-AG) (Sugiura et al, 1995), which were detected in samples of the brain and intestine and shown to activate $CB_1$ and $CB_2$ with high affinity and efficacy, were subsequently identified as endocannabinoids (Di Marzo & Fontana, 1995; Cristino et al, 2020). The endocannabinoid system can be further expanded to the endocannabinoidome, a much wider complex network of promiscuous mediators overlapping with other signaling pathways, including LPA and its receptors (Cristino et al, 2020). Interestingly, it has been shown that dephosphorylation of a 2-arachidonoyl species of LPA in the brain of rats leads to the formation of the endo-cannabinoid 2-AG (Nakane et al, 2002), a process that was later revealed to depend on lipid phosphate phosphatases (Aaltonen et al, 2012). In addition, the two endocannabinoid receptors, $CB_1$ and $CB_2$, show an amino acid sequence identity to $LPA_{1-3}$ of around 18–20% (Chun et al, 1999). Moreover, a functional crosstalk between $CB_1$ and $LPA_1$ has been revealed, where the absence of the main cerebral receptors for LPA or endocannabinoids is able to induce a modulation on the other at the levels of both signaling and synthesis of endogenous neurotransmitters (González de San Román et al, 2019).

Pharmacological manipulation of the endocannabinoid system can be achieved by medicinal cannabis. The major psychoactive cannabinoid component of medicinal cannabis from the plant *Cannabis sativa* is $\Delta^9$-*trans*-tetrahydrocannabinol (THC), which can bind to $CB_1$ and $CB_2$ in a low nanomolar regime (Pertwee, 2008). Here, we show the potential of THC, and other cannabinoid compounds, to modulate the catalytic activity of ATX, and present results that THC can reduce ATX-mediated LPA signaling in a cellular context.

## Results and Discussion

### Inhibition of ATX by various cannabinoids

We first set out to validate the hypothesis that cannabinoids might bind ATX to modulate its catalytic function. For this, we used various cannabinoids (Fig 1A) at a fixed concentration of 10 $\mu$M for each

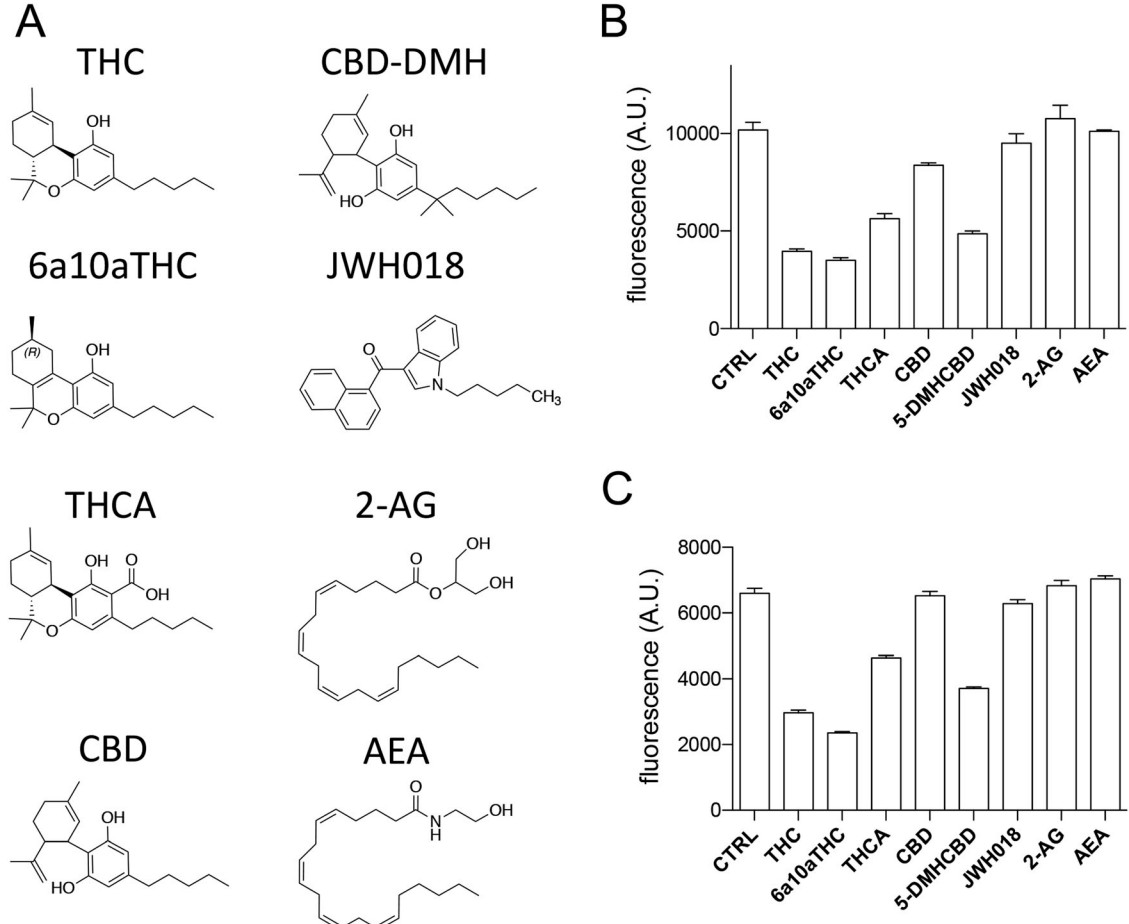

**Figure 1.  End-point assays of compounds tested.**
**(A)** Chemical representation of $\Delta^9$-*trans*-tetrahydrocannabinol (THC), cannabidiol-dimethylheptyl (CBD-DMH), 9(R)-$\Delta$6a,10a-THC (6a10aTHC), JWH018, tetrahydrocannabinolic acid, 2-arachidonoylglycerol, cannabidiol (CBD), and anandamide. **(B, C)** End-point assay for (B) ATX-$\beta$ and (C) ATX-$\gamma$ inhibition with various cannabinoids and endocannabinoids. All error bars represent the SEM (n = 3). **(B, C)** Activity rate of 99 and 65 mMol end product/mM ATX/min has been found for (B, C), respectively. ANOVA comparison between CTRL and other conditions showed statistically significant differences for THC, 6a10aTHC, tetrahydrocannabinolic acid, CBD, and 5-DMH-CBD for ATX-$\beta$ and ATX-$\gamma$ inhibition ($P < 0.005$). ATX was not significantly inhibited by JWH018, anandamide, and 2-arachidonoylglycerol ($P > 0.005$).

compound in our biochemical validation with LPC 18:1 (200 $\mu M$) as substrates in an end-point assay for ATX-$\beta$ (UniProt ID: Q13822-1) and ATX-$\gamma$ (UniProt ID: Q13822-3) (Fig 1B and C). The quality of our enzyme assay is confirmed with HA155, a well-documented ATX inhibitor (Fig S1A) (Albers et al, 2011; Hausmann et al, 2011). The obtained $IC_{50}$ (6 ± 0.8 nM) for HA155 is similar in our assay conditions compared with previous results (Albers et al, 2011). In addition, we can exclude interference of the cannabinoid compounds in our enzymatically coupled assay, as no difference is detectable in the absence of ATX and LPC (Fig S1B), performed in the presence of choline.

We observe a potent inhibition of THC on the catalysis of both ATX isoforms with more than 50% inhibition (Fig 1B and C). Furthermore, 9(R)-Δ6a,10a-THC (for simplicity referred to from here as 6a10aTHC), a derivative of THC that differs only in the position of the double bond in the C-ring compared with THC (Fig 1A), is included in our cannabinoid inhibition screen. Interestingly, this minimal difference causes a further increase in the magnitude of inhibition for both ATX isoforms tested (Fig 1B and C). Tetrahydrocannabinolic acid is a precursor of THC and an active component of medicinal cannabis. It is distinguishable from THC by the presence of a carboxylic group at the A-ring (Fig 1A). Tetrahydrocannabinolic acid also showed an inhibition of the enzymatic activity of both ATX isoforms tested. However, this inhibition is less pronounced, when compared to THC and 6a10aTHC, and did not reach a 50% inhibition magnitude in our assay conditions (Fig 1B and C).

The next compound we tested was cannabidiol (CBD), a non-psychoactive ingredient of medicinal cannabis. CBD is structurally different from THC by an opening of the B-ring. Interestingly, CBD showed only a weak inhibition toward ATX-$\beta$, and no observable inhibition for ATX-$\gamma$ (Fig 1B and C). CDB-DMH is a synthetic CBD derivative, which is characterized by the addition of two methyl groups at the beginning of the aliphatic chain and an elongation with a single methyl group at the end of the carbon chain (Fig 1A). These structural modifications remarkably increase the magnitude of inhibition for ATX-$\beta$ and ATX-$\gamma$ (Fig 1B and C).

We also analyzed JWH018 (Fig 1A), which is a synthesized compound and full agonist for $CB_1$ and $CB_2$ with $K_i$ values of 9.0 ± 8.0 and 2.9 ± 2.7 nM, respectively (Aung et al, 2000). However, this artificial cannabinoid did not influence the catalytic activity of either ATX-$\beta$ or ATX-$\gamma$ isoforms (Fig 1B and C). To complete our picture of the modulation cannabinoids on the enzymatic activity of ATX, we also used the endocannabinoids 2-AG and anandamide. However, both endocannabinoids did not affect the catalysis of ATX in the applied conditions (Fig 1B and C).

### Biochemical characterization of THC and 6a10aTHC with ATX-$\beta$ and ATX-$\gamma$

We choose THC and 6a10aTHC for our detailed biochemical characterization, as these inhibitors have a maximum magnitude of inhibition of more than 50%, a cutoff criterion selected under the assay conditions used. THC works as a partial inhibitor on the catalysis of both isoforms (Fig 2A and B). The apparent $EC_{50}$ value of THC with ATX-$\beta$ and LPC 18:1 as a substrate is 1,026 ± 138 nM, as shown in Fig 2A. The magnitude of inhibition is around 60%. A

similar magnitude of inhibition is observed with ATX-$\gamma$, with an apparent $EC_{50}$ of 407 ± 67 nM for THC toward this isoform (Fig 2B).

Next, we validate the artificial THC derivative 6a10aTHC. The apparent $EC_{50}$ value of 6a10aTHC for ATX-$\beta$ is 844 ± 178 nM (Fig 2C) and thus comparable to THC. The maximum inhibition is marginally increased and appears to be around 75%. 6a10aTHC has the highest potency toward ATX-$\gamma$ with a determined apparent $EC_{50}$ of 374 ± 66 nM (Fig 2D). The magnitude of inhibition is around 70%, which is consistent with ATX-$\beta$. Overall, 6a10aTHC is the best used cannabinoid inhibitor for both isoforms tested with the classical substrate LPC 18:1, and also with LPC 16:0 (Fig S2).

Both THC and 6a10aTHC are very closely related in structure to each other and show a similar behavior in biochemical assays. Thus, we performed a mode of inhibition analysis with THC only, to understand the inhibition mode of these compounds (Fig 2E). This analysis is carried out with 0, 0.35, 0.7, and 1.4 $\mu M$ of THC with geometrically increasing concentrations of LPC 18:1. It revealed that THC functions as a mixed-type inhibitor, which is demonstrated by the decrease in $V_{max}$ from 8.5 to 7.3, 6.3, and 5.0, respectively, and an increase in $K_m$ from 10.1 to 19.6, 29.9, and 31.5 $\mu M$, respectively.

### Co-crystal structure of ATX-THC

To understand the binding interface between THC and ATX in detail, we expressed and purified the second isoform of ATX from *Rattus norvegicus* (UniProt ID: Q64610-2, rATX-$\beta$) and co-crystallized this formerly used ATX construct (Hausmann et al, 2011; Keune et al, 2016) with THC. We determined this ATX-THC structure (PDB ID: 7P4J) to 1.8 Å resolution with an $R_{free}$ of 23.5% (Table 1). We obtained clear residual electron density close to the active site of ATX. Modeling of THC here resulted in a very good fit to this remaining electron density (Fig 3A). ATX binds to THC at the entrance of the hydrophobic pocket with the aliphatic chain pointing into this pocket. The binding of the THC molecule is driven by hydrophobic interactions of the residues I167, F210, L213, L216, W254, F274, Y306, and V365 (Fig 3B), as analyzed by the PLIP server (Adasme et al, 2021). A superposition of our ATX-THC structure with the ATX-LPA 18:1 structure (PDB ID: 5DLW) (Keune et al, 2016) shows that the THC molecule blocks binding of the LPA 18:1 aliphatic chain, whereas binding to the glycerol backbone and the phosphate group can still occur (Fig 3C).

### Co-crystal structure of ATX-6a10aTHC

We also obtained an ATX-6a10aTHC structure (PDB ID: 7P4O) to 1.7 Å resolution with an $R_{free}$ of 20.6% (Table 1). In this ATX-6a10aTHC structure, we observed clear residual electron density, which resembles almost perfectly the 6a10aTHC ligand (Fig 4A). The binding of the 6a10aTHC molecule is again mainly accomplished by hydrophobic interactions of the residues I167, F210, L213, W254, F273, F274, and Y306 (Fig 4B), as analyzed by the PLIP server (Adasme et al, 2021). Nevertheless, an additional water bridge between the carbonyl of F273 and the THC derivative can be observed (Fig 4C), which suggests that the binding stability of this ligand is higher compared with THC, and potentially explains the lower apparent $EC_{50}$ for 6a10aTHC. However, the authors are aware that a comparable water

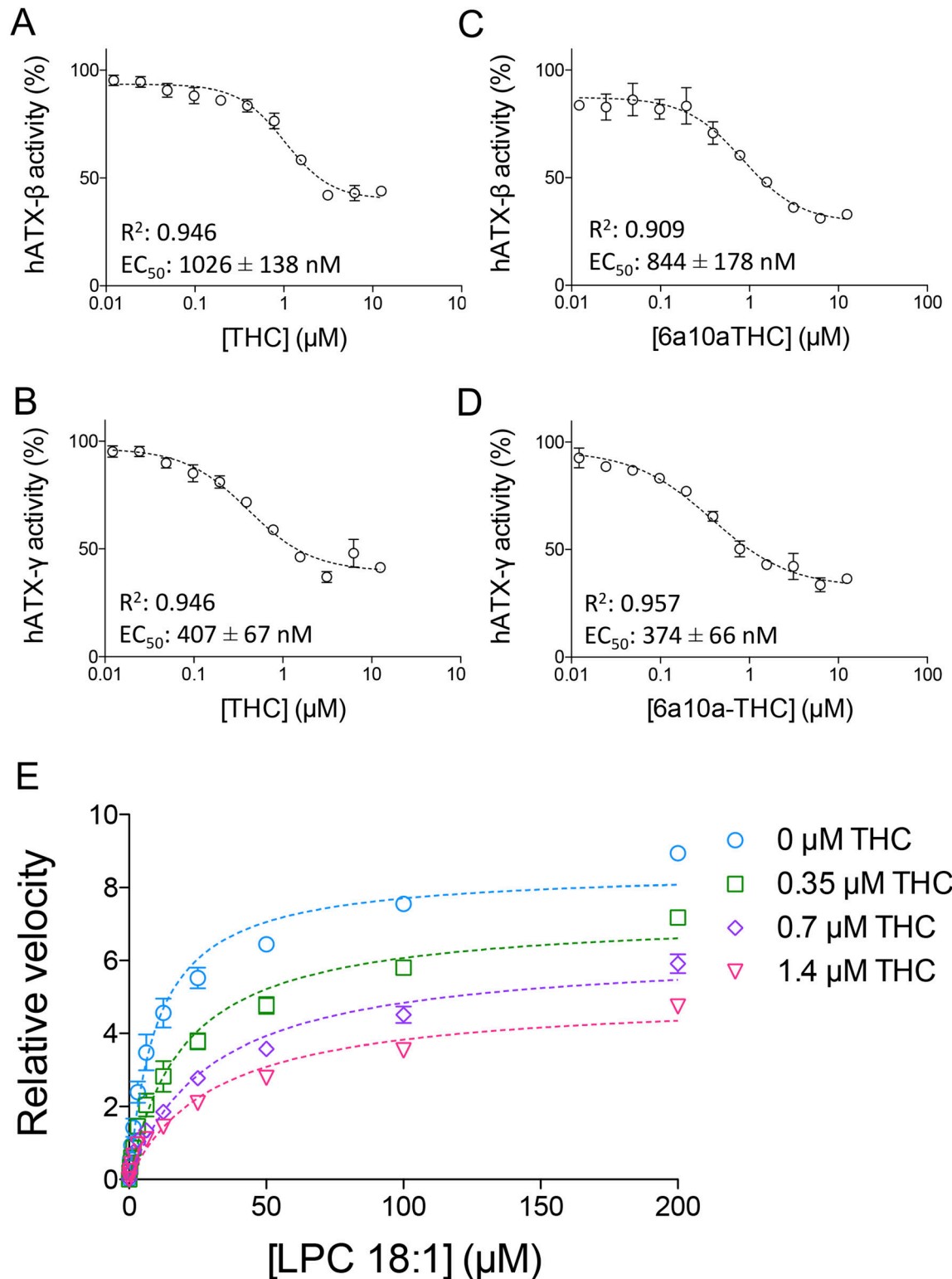

**Figure 2. Inhibition of ATX by plant-derived THC and synthetic 6a10aTHC.**
**(A, B, C, D)** Dose–response analysis of (A) ATX-$\beta$ and (B) ATX-$\gamma$ with THC and LPC 18:1, and of (C) ATX-$\beta$ and (D) ATX-$\gamma$ with 6a10aTHC and LPC 18:1. **(E)** Mode of inhibition of THC with ATX-$\gamma$ indicates a mixed-type inhibition. All error bars represent the SEM (n = 3).

**Table 1. Crystallographic data collection and refinement statistics.**

| Crystal | ATX-THC | ATX-9(R)-Δ6a,10a-THC |
|---|---|---|
| PDB identifier | 7P4J | 7P4O |
| Data collection | | |
| Wavelength (Å) | 0.976 | 1.000 |
| Space group | P1 | P1 |
| Cell dimensions | | |
| a, b, c (Å) | 53.7 61.0 63.6 | 53.8 62.4 64.4 |
| α, β, γ (°) | 103.2 97.4 94.2 | 103.7 98.4 93.4 |
| Resolution (Å)[a] | 61.2-1.8 (1.9-1.8) | 53.0-1.7 (1.75-1.7) |
| No. of reflections | 47,909 (2,395) | 85,204 (8,434) |
| $R_{pim}$ (%) | 5.8 (64) | 7.25 (63.8) |
| Completeness (%) | | |
| Spherical | 65.9 (13.1) | 94.8 (94.2) |
| Ellipsoidal | 91.6 (60.4) | — |
| Redundancy | 9.2 (7.0) | 3.5 (3.6) |
| Refinement | | |
| $R_{work}$ (%) | 18.69(29.1) | 17.10 (21.5) |
| $R_{free}$ (%) | 23.5(22.8) | 20.60 (25.3) |
| No. of atoms[b] | 6,800 | 6,818 |
| Protein + carbohydrates | 6,230 | 6,244 |
| Ligand + metal ions | 163 | 113 |
| Waters and other ions | 407 | 461 |
| B-factors (Å²) | | |
| All | 27.5 | 31.9 |
| Protein + carbohydrates | 27.0 | 31.3 |
| Ligand + metal ions | 34.3 | 42.1 |
| Water and other ions | 32.9 | 36.6 |
| R.m.s. deviations | | |
| Bond lengths (RMS) | 0.005 | 0.007 |
| Bond angles (RMS) | 0.86 | 0.90 |

[a]Values given in parenthesis refer to reflections in the highest resolution bin. For calculation of $R_{free}$, 5% of all reflections were omitted from refinement. [b]Alternate conformations are counted as multiple atoms.

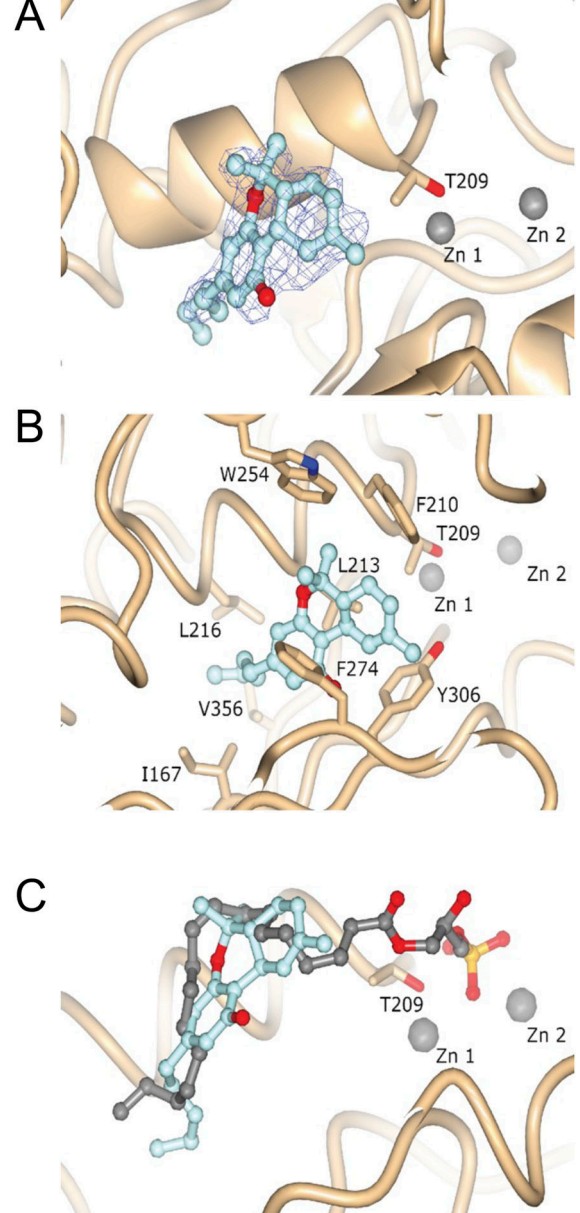

**Figure 3. Structure and electron density of ATX-THC.**
**(A)** Feature-enhanced electron density map before THC placement, contoured at 1 RMSD and represented as a blue wireframe model. **(B)** Molecular interactions of THC with ATX. **(C)** Superposition of ATX binding to THC (PDB ID: 7P4J) and LPA 18:1 (PDB ID: 5DLW).

molecule also exists in the ATX-THC structure, where the distance between the THC oxygen and carbonyl oxygen of F273 is 4.5 Å, thus above the PLIP server threshold for such an interaction (Fig S3).

The 6a10aTHC ligand in ATX adopts an overall similar binding position to the cannabinoid in our ATX-THC structure. However, the aliphatic chain of the ligand points in a slightly different direction when compared to THC. Also noteworthy is that the cyclohexene (C-ring) appears to adopt a different stereoisomeric configuration (Fig S4) because of the alternate localization of the double bond (Fig 1A).

## Inhibition of LPA₁ internalization in HeLa cells by THC

To validate THC can act as an inhibitor in the production of LPA and thus ATX-LPA signaling in a cellular context, we used an agonist-induced LPA₁ receptor internalization as a readout in cultured cell assays (Murph et al, 2003; Lee et al, 2006, 2007). As shown in Fig 5, stimulation of HA/LPA₁-transfected HeLa cells with 30 nM ATX, 150 $\mu$M LPC 18:1, and 1 $\mu$M THC significantly reduced LPA₁ internalization. This observation was only detectable in the presence of ATX and not in control conditions (Fig S5). This is an indirect response to blocking LPA production, which inhibits receptor activation and endocytosis, confirming a more physiological role of THC as a potent inhibitor.

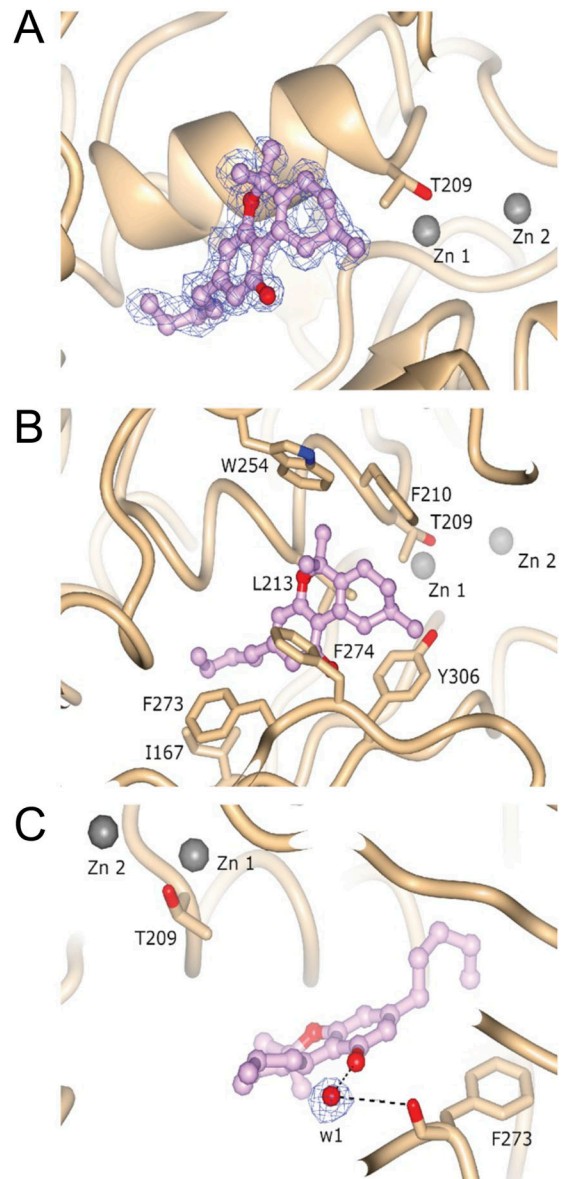

**Figure 4. Structure and electron density of ATX-6a10aTHC.**
**(A)** Featured-enhanced electron density map before 6a10aTHC placement, contoured at 1 RMSD and represented as a blue wireframe model. **(B)** Molecular interactions of 6a10aTHC with ATX. **(C)** Bridging water molecule interaction between 6a10aTHC and carbonyl oxygen of F273.

## Discussion

Medicinal cannabis has been approved as a therapeutic agent by local authorities in an increasing number of states all over the world. Even though great progress in the molecular basis of medicinal cannabis therapy has been achieved over the last decades, the pleiotropic effects have been insufficiently characterized to date. We establish here a potent in vitro inhibition of various cannabinoids, such as THC, on the catalysis of ATX with different substrates (LPC 16:0 and LPC 18:1) and isoforms. Based on our investigations, we provide evidence that THC can potently modulate LPA signaling.

In most studies that try to address the pharmacological aspects of medicinal cannabis, the administration has been performed via smoking. In this context, the first body fluid that encounters THC is the saliva. The mean concentration of THC in this oral fluid has been detected with up to 4,167 ng/ml (13 $\mu$M) in a radioimmunoassay (Huestis & Cone, 2004). Interestingly, LPA is present in saliva (Sugiura et al, 2002) and ATX expression can be detected in salivary gland tissue (Maruya et al, 2004), suggesting that ATX-LPA signaling may be reduced by THC in vivo. Furthermore, the deposition of THC in oral fluid reflects a similar time course in plasma after smoked cannabis administration (Huestis & Cone, 2004). Serum concentrations of THC show a wide inter-individual difference, between 59 and 421 ng/ml after a 49.1-mg THC dose, which equals 190 nM to 1.3 $\mu$M of THC (Hunault et al, 2008). The observed mean THC peak in this study for a 69.4-mg THC dose was 190.4 ng/ml (SD = 106.8), which is in the range of the apparent $EC_{50}$ determined during our studies.

95–97% of THC is bound to plasma proteins, and our data suggest that ATX might function as a carrier for THC in plasma. The authors extensively tried to determine the binding affinity of ATX for THC with various techniques, such as isothermal titration calorimetry and nuclear magnetic resonance; however, these approaches were unsuccessful because of the hydrophobic nature of the cannabinoid ligand.

Recently, the potential use of medicinal cannabis in tissue fibrosis has been proposed (Pryimak et al, 2021). In this regard, it is noteworthy to mention the role of ATX in idiopathic pulmonary fibrosis. Several inhibitors targeting ATX are under clinical investigations for their therapeutic use against idiopathic pulmonary fibrosis (Zulfikar et al, 2020). However, the ISABELA study (clinical phase 3 investigation) of the most advanced molecule targeting ATX, Ziritaxestat (Glpg1690) from Galapagos N.V, was discontinued because of risk–benefit concerns. It is tempting to speculate that a full ATX inhibitor, which reduces LPA levels to almost zero, causes many systemic unwanted side effects, as the ATX-LPA signaling axis is pivotal under physiological conditions. In this context, our observation that THC is a partial inhibitor of ATX is of great interest, because this molecule is an FDA-approved drug, which could reduce LPA levels incompletely. Moreover, the fact that THC can cross the blood–brain barrier makes it an attractive candidate to manipulate neuronal diseases, where the brain-specific isoform of ATX is involved.

In addition, glaucoma is the leading cause of irreversible blindness worldwide (Tham et al, 2014). Glaucoma is characterized by elevated intraocular pressure levels, and medicinal cannabis is being used to treat this pathology; however, the therapeutic mechanism is not completely known. Interestingly, in recent years it has been discovered that aqueous humor samples of patients suffering from primary open-angle glaucoma have elevated levels of ATX, LPC, and LPA (Ho et al, 2020). Moreover, pharmacological inhibition of ATX lowered intraocular pressure in rabbits (Iyer et al, 2012). Our data may explain the molecular basis for the therapeutic effect of medical cannabis in glaucoma patients, as THC could feasibly reduce the formation of LPA by inhibiting the enzymatic activity of ATX.

In conclusion, our study warrants further research into the pleiotropic effects of medicinal cannabis in the context of ATX-LPA

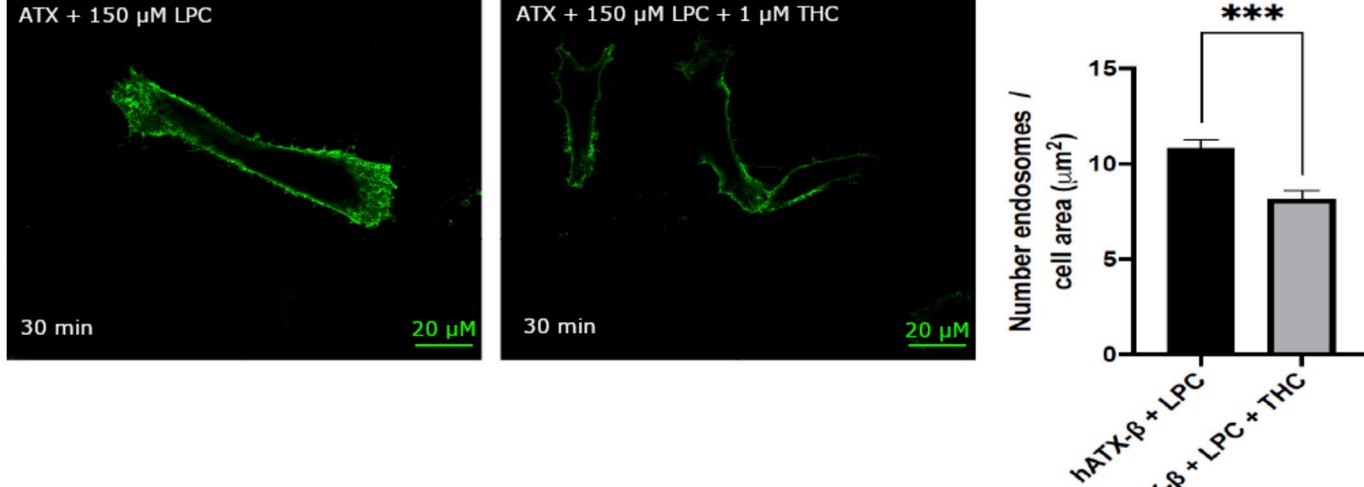

**Figure 5. Physiological effect of THC on LPA$_1$ receptor internalization.**
Quantification of LPA$_1$ receptor internalization revealed that THC reduced the number of endosomes internalized when compared to untreated condition, paired $t$ test, $P = 0.0008$. All error bars represent the SEM, calculated from 11 images per condition in biological triplicate experiments.

signaling, while also providing a promising starting point for such research lines. Furthermore, this work also provides a scaffold for the design of new inhibitors for further studies of the ATX-LPA signaling axis, and suggests a new way to intervene in ATX-LPA signaling–mediated pathologies with THC.

# Materials and Methods

## Materials

We obtained T300 tissue culture flasks (#90301) from TPP and roller bottles (#681070) from Greiner Bio-One; DMEM (#12491023; Gibco), Opti-MEM (#31985062; Gibco), FBS (#10270106; Gibco), fatty acid–free FBS (#A3382101; Gibco), L-glutamine (#25030-123; Gibco), POROS-20 MC column (#1542906; Thermo Fisher Scientific), Lipofectamine 3000 (#L3000001; Thermo Fisher Scientific), Alexa Fluor 594 conjugate (#W11262; Invitrogen), and SDS precast gel (#XP04205BOX; Invitrogen) from Thermo Fisher Scientific; CELLSTAR 12-well culture plates (#665180; Greiner) and Fluoroshield (#F6182-20 ml) from Sigma-Aldrich; Amicon ultra 15-ml 10-kD (#UFC901008) and ultra 0.5-ml 10-kD (#UFC501008) concentrators from Merck; Superose 6 (10/300) column (#17-5172-01) and Superdex 200 Increase 3.2/300 column (#28-9909-46) from GE Healthcare; Trans-Blot Turbo transfer pack (#1704158) from Bio-Rad; LPC 18:1 (#845875P), LPC 16:0 (#855675C), and LPA 18:1 (#857130C) from Avanti Polar Lipids; choline quantification kits (#40007) from AAT Bioquest; THC (#LGCAMP1088.00-05) from LGC Standards, France; 9(R)-Δ6a,10a-THC (#33013) from Cayman; CBN (#C-046-1ML) from Cerilliant; 5-DMH-CBD (#1481) from Tocris; CBD (#HB2785) from HelloBio; NH$_4$I (#AB202711) from Abcr; NaSCN (#HR2-693) from Hampton Research; and InstantBlue Coomassie protein stain (#ab119211), anti-HA tag primary antibody (#ab18181), anti-mouse antibody (#ab150113), anti-ATX antibody (#ab77104), anti-mouse HRP secondary antibody (#ab6728), and ECL substrate kit (#ab133406) from Abcam.

## Methods

### ATX expression and purification

Recombinant ATX proteins were essentially produced as previously described (Hausmann et al, 2010). HEK293-Flp-In cells were cultivated in complete DMEM supplemented with 10% FBS with minor differences. The cells were grown to 80–90% confluence, washed twice with preheated PBS, and trypsinized for 5 min with 5 ml of trypsin. Inactivation was accomplished by adding 45 ml of complete medium. Cells were resuspended in complete medium and inoculated into roller bottles. 10 T300 flasks were used to inoculate eight roller bottles, and the cells were cultured for 4 d after transfer into 125 ml DMEM containing 10% FBS and 2 mM glutamine. The medium was then replaced by 125 ml DMEM containing 2% FBS and 2 mM glutamine. The cells were then left to express protein for 4–5 d before collection. Fresh expression medium was added for a further round of recombinant expression.

HEK293 medium from eight roller bottles was pooled together, and the recombinant ATX proteins were purified using a POROS-20 MC column preloaded with Cu$^{2+}$. Equilibration was achieved by washing with 10 column volumes of buffer A (20 mM Hepes and 150 mM NaCl, pH 7.4). ATX was eluted with a linear gradient of buffer B consisting of buffer A supplemented with 500 mM imidazole. Reasonably, pure fractions were pooled after SDS–PAGE analysis. The fraction volume was reduced with an Amicon ultra 15-ml 10-kD concentrator to a volume of 500 $\mu$l. 5 mg/ml concentrated protein was injected into a Superose 6 (10/300) gel filtration column using buffer A. The purity of the peak fractions was analyzed by SDS–PAGE. The recombinant protein concentration was determined by the ratio of the optical density at 260/280 nm using a NanoDrop 2000

spectrophotometer (Thermo Fisher Scientific). The ATX construct from *Rattus Norvegicus* (UniProt ID: Q64610-2) was concentrated to 3–3.5 mg/ml using an Amicon ultra 0.5-ml 10-kD concentrator. *Human* ATX (UniProt ID: Q13822-1 and Q13822-3) was concentrated to 1.3 mg/ml. Purity was assessed using SDS–PAGE, Western blot, and SEC analysis (Fig S6A–C). For the SDS–PAGE, 25 µg of purified protein in reducing conditions was loaded on a precast gel, run for 1 h at 225 V, and imaged after InstantBlue Coomassie protein staining following the manufacturer's instructions. For Western blot, proteins were transferred using the Trans-Blot Turbo transfer system (Bio-Rad). Membrane staining with the primary antibody was performed overnight using an anti-ATX antibody. Anti-mouse HRP secondary antibody was incubated for 1 h, and detection was then performed using an ECL substrate kit. Analytical SEC was performed by injecting 25 µl of rATX on a Superdex 200 Increase 3.2/300 column equilibrated with a 50 mM Tris–HCL (pH 8) and 150 mM NaCl buffer. hATX-$\beta$ and hATX-$\gamma$ activity was controlled using the choline release assay described below (Fig S6D). hATX-$\gamma$ activity was slightly lower than hATX-$\beta$ activity, which is in accordance with published comparison of the different ATX isoforms (Giganti et al, 2008). ATX-$\beta$ and ATX-$\gamma$ activity was monitored in the presence and absence of 0.5 mg/ml albumin (Fig S6E and F).

### End-point assays

ATX lysophospholipase D activity was measured using choline release from LPC 18:1 and LPC 16:0 with a choline quantification kit (Hausmann et al, 2016) (Fig S6D). 30 nM ATX-$\beta$ or ATX-$\gamma$ was incubated with 200 µM LPC 18:1 or LPC 16:0 in a final volume of 100 µl buffer, which contained 50 mM Tris–HCl (pH 8.5) and 150 mM NaCl. The LPC solution was obtained by evaporating a commercial LPC chloroform solution directly in a 1.5-ml Eppendorf tube under vacuum. The dried LPC was then resuspended in water to obtain the mentioned concentration and incubated at 37°C on a shaker for 5–10 min before addition to the microplate. The cannabinoid solutions were prepared from a stock solution in ethanol or other organic solvents. After initial evaporation and/or dilution to obtain the highest concentration mentioned in the assay figure, a twofold dilution was performed in Eppendorf tubes. All the compounds were dissolved in 100% ethanol as a vehicle.

The experiments for determining relative inhibition for various cannabinoids were performed at 37°C by adding 10 µM of the cannabinoid or endocannabinoid mentioned. Released choline was detected, and the enzyme activity was determined by measuring fluorescence at $\lambda_{ex}/\lambda_{em}$ = 540/590 nm in 96-well plates, every 60 s for 50 min minimum using a CLARIOstar plate reader (BMG Labtech). Absolute values were taken at 25 min after visual inspection, and the 0-min baselines were subtracted to account for compound differences. The relative inhibition values were determined using the normalize method in GraphPad Prism (GraphPad Software, Inc.). Measurements have been performed in triplicate with three different protein preparations. All the compounds were controlled for interference of fluorescence and inhibition in the same assay conditions but in the absence of ATX and replacing LPC with choline.

### Dose–response assay for cannabinoids

ATX lysophospholipase D activity was measured using choline release from LPC 18:1 and LPC 16:0 with a choline quantification kit

(Hausmann et al, 2016). 30 nM ATX-$\beta$ or ATX-$\gamma$ was incubated with 200 µM LPC 18:1 or LPC 16:0 in a final volume of 100 µl buffer, which contained 50 mM Tris–HCl (pH 8.5) and 150 mM NaCl. The LPC solution was obtained by evaporating a commercial LPC chloroform solution directly in a 1.5-ml Eppendorf tube under vacuum. The dried LPC was then resuspended in water to obtain the mentioned concentration and incubated at 37°C on a shaker for 5–10 min before addition to the microplate.

The experiment for determining apparent $EC_{50}$ for various cannabinoids was performed at 37°C by adding the cannabinoid in a serial twofold dilution for each concentration. The cannabinoid solutions were prepared from a stock solution in ethanol or other organic solvents. After initial evaporation and/or dilution to obtain the highest concentration mentioned in the assay figure, a twofold dilution was performed in Eppendorf tubes. All the compounds were dissolved in 100% ethanol as a vehicle. For THC and 6a10aTHC, the retained twofold dilution started at 12.5 µM. For CBN, 5-DMH-CBD, and CBD, the starting concentration was 50 µM, 150 µM, and 2 mM, respectively. Released choline was detected, and the enzyme activity was determined by measuring fluorescence at $\lambda_{ex}/\lambda_{em}$ = 540/590 nm in 96-well plates, every 60 s for 50 min minimum using a CLARIOstar plate reader (BMG Labtech). Initial velocities were taken between 19 and 31 min after visual inspection. The apparent $EC_{50}$ values were determined using the non-linear regression analysis method (fit: [inhibitor] versus response [three parameter]) in GraphPad Prism (GraphPad Software, Inc.). However, it should be mentioned that the relative concentrations of the different lipids/inhibitors in their physical form as micelles, liposomes, protein-bound, or aggregates are unknown (Carman et al, 1995). These uncertainties are widely known, and we provide here an apparent $EC_{50}$ for consideration. Measurements have been performed in triplicate with three different protein preparations.

### Biochemical data analysis

The data analysis was performed with GraphPad (9.4.1). For apparent $EC_{50}$ determination, fluorescent time points are subtracted from the baseline. From the subtracted results, a linear regression analysis was run on the linear part of the fluorescent curve, between 10 min and 25 min. The linear regression slopes were then plotted and normalized for each inhibitor concentration. A non-linear regression analysis using the following equation was performed with GraphPad to calculate the apparent $EC_{50}$:

$$Y = \text{Bottom} + (\text{top} - \text{bottom}) \Big/ \left(1 + [EC50/X]^{\text{hillslope}}\right).$$

The apparent $EC_{50}$ was calculated as the concentration of inhibitor that gives a response halfway between maximal and minimal ATX activity. The S.E.M. of the apparent $EC_{50}$ was determined by GraphPad Prism as the 95% confidence interval of the mean.

### Choline standard

The assay was run as mentioned in the previous dose–response material and methods above, apart from the replacement of LPC by the choline standard, as mentioned in the manufacturer's instructions. The obtained curve is linear allowing extrapolation of the enzyme activity.

### Crystallization, structure determination, and model building

Crystallization experiments were performed at 303 K using the hanging-drop vapor diffusion method as previously published (Day et al, 2010). The best crystals were obtained with the *r*ATX construct (3–3.5 mg/ml) after 30-min RT preincubation with 5 mM THC or 5 mM 6a10aTHC dissolved in ethanol. 1 $\mu$l of the protein solution was then mixed with 1 $\mu$l of the reservoir solution containing 18–22% (m/v) PEG3350, 0.1–0.3 M $NH_4I$, and 0.3 M NaSCN. All the crystals were cryoprotected with the addition of 20% (vol/vol) glycerol.

X-ray data for THC and 6a10aTHC ATX complexes were collected at 100 K on EMBL PETRA III beamlines P14 and P13 (Cianci et al, 2017), respectively. Crystallographic ATX-THC complex data were acquired using the Global Phasing WFs data collection workflow to maximize the completeness of the P1 dataset. Authorization to collect sample containing THC was granted by the BfArM in Germany. All data were processed with autoPROC (Vonrhein et al, 2011)/STARANISO, which includes XDS (Kabsch, 2010). Structures were determined by molecular replacement using MRage (Adams et al, 2010) with the structure of ATX (PDB: 2XR9) as a model (Hausmann et al, 2011). Model building was performed using Coot (Emsley & Cowtan, 2004), phenix.refine (Afonine et al, 2012), REFMAC5 (Murshudov et al, 2011), and PDB-REDO (Joosten et al, 2009). Ligands were drawn with ELBOW (Moriarty et al, 2009). Validation of the model was performed with phenix PDB deposition tools, using MolProbity (Williams et al, 2018). Maps were generated using phenix.refine and feature-enhanced map (Afonine et al, 2015). The crystallographic parameters and model quality indicators are shown in Table 1. Structural figures were generated using CCP4mg (McNicholas et al, 2011). Structural biology applications used in this project were compiled and configured by SBGrid (Morin et al, 2013).

### hLPA1 receptor internalization assay

The hLPA$_1$ receptor internalization assay was essentially performed as previously described (Lee et al, 2007). A pRP[Exp]-Puro-CMV > HA/hLPA$_1$ vector coding for full-length human LPA$_1$ receptor (UniProt ID: Q92633) with a human influenza HA sequence epitope tag at the 5′-end of the extracellular domain was designed, and maxiprep plasmid DNA was produced commercially (VectorBuilder). Vector quality control was done by restriction enzyme analysis and Sanger sequencing.

HeLa cells were grown on coverslips in a 12-well plate format and transfected with HA/hLPA$_1$ vector in DMEM complete medium with Lipofectamine 3000 using 1 $\mu$g of plasmid DNA, and 3 $\mu$l of Lipofectamine 3000 per well after complexation in 50 $\mu$l Opti-MEM, as per the manufacturer's instructions, 48 h before fixation. 8 h before treatment and fixation, the cells were starved in fatty acid–free DMEM to avoid hLPA$_1$ activation by serum lipids. Several assays were performed in different conditions before fixation: 30 nM ATX + 150 $\mu$M LPC 18:1; 30 nM ATX + 150 $\mu$M LPC 18:1 + 1 $\mu$M THC; 1 $\mu$M THC; 1 $\mu$M LPA 18:1; untreated (vehicle only); and untransfected, to control specificity of the antibody towards HA-tagged hLPA$_1$ receptor. LPC 18:1 and LPA 18:1 were dissolved in fatty acid–free FBS with a final concentration in the media of 1%. THC was dissolved in DMSO to a final concentration in the media of 0.025% (vol/vol) DMSO.

Fixation was carried out by adding paraformaldehyde directly into the media to a final concentration of 3%, and incubating at 37°C for 10 min. Cells were washed three times in PBS, and membranes were labeled using wheat germ agglutinin, and Alexa Fluor 594 conjugate for 10 min at 5 $\mu$g/ml in PBS, as per the manufacturer's instructions. Cells were washed three times in PBS, permeabilized using 0.2% Tween for 10 min, washed in PBS, and finally blocked with 10% goat serum for 30 min. HA tag was labeled using an anti-HA tag primary antibody at 1/200 dilution in 10% FBS for 1 h at room temperature followed by PBS wash, and secondary staining was done with an anti-mouse antibody, with 30-min incubation at 1/500 dilution in 10% FBS. Cells were washed three times and mounted using Fluoroshield mounting medium. Imaging was performed using a Leica SP5 (× 63 objective). Endosome quantification was done using Fiji Analyze Particle tools after image thresholding. The number of counted endosomes was normalized over the measuring area to calculate the density per $\mu m^2$. Statistical analysis was performed using a paired *t* test over 11 images for each condition of ATX-THC-LPC and ATX-LPC in biological triplicate.

## Supplementary Information

## Acknowledgements

The authors are grateful for the initial in silico work of Dr. Ulrike Uhrig of the Chemical Biology Core Facility, EMBL, who demonstrated a potential inhibition of THC towards ATX, which was fundamental to apply for an initial THC license. The authors are also grateful to Dr. Corinna Gorny for her support to obtain a license for Dronabinol experiments at EMBL Germany under BfArM authorization. We are thankful for the kind gift of ATX-expressing cell lines from the Perrakis laboratory in Amsterdam. We thank beamline staff from the EMBL-Hamburg beamlines at the PETRA III storage ring (DESY, Hamburg, Germany) for beamtime (proposal number MX-661). In particular, we thank Drs. G Bourenkov, and G Bricogne and R Fogh at Global Phasing Limited who provided valuable assistance with multi-orientation data collection strategies on P14. The authors also wish to express their gratitude to the Eukaryotic Expression Facility in Grenoble for infrastructure access, especially Alice Aubert and Martin Pelosse for excellent technical support and fruitful discussions. Lastly, we thank L Gutierrez for her technical support with biochemical assays. MC Eymery has been funded by the EMBL International PhD program. AA McCarthy has been funded by EMBL. J Hausmann was supported by a fellowship from the EMBL Interdisciplinary Postdoc (EI3POD) program under Marie Skłodowska-Curie Actions COFUND (grant number: 664726).

### Author Contributions

MC Eymery: data curation, formal analysis, investigation, visualization, methodology, and writing—review and editing.
AA McCarthy: resources, supervision, funding acquisition, validation, project administration, and writing—review and editing.
J Hausmann: conceptualization, resources, formal analysis, supervision, validation, and writing—original draft.

### Conflict of Interest Statement

The authors declare that they have no conflict of interest.

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
