## [Reviewer comments · Life Science Alliance]

Life Science Alliance

Linking medicinal cannabis to autotaxin-lysophosphatidic acid signaling.

Mathias Eymery, Andrew McCarthy, and Jens Hausmann

DOI: <https://doi.org/10.26508/lsa.202201595>

Corresponding author(s): Andrew McCarthy, European Molecular Biology Laboratory and Jens Hausmann, European Molecular Biology Laboratory

Review Timeline:

Submission Date:	2022-07-08
Editorial Decision:	2022-07-31
Revision Received:	2022-11-07
Editorial Decision:	2022-11-22
Revision Received:	2022-11-25
Accepted:	2022-11-25

Scientific Editor: Novella Guidi

Transaction Report:

July 31, 2022

Re: Life Science Alliance manuscript #LSA-2022-01595-T

Dr. Andrew A. McCarthy
European Molecular Biology Laboratory
Grenoble
71 avenue des Martyrs
Grenoble Cedex 9 38042
France

Dear Dr. McCarthy,

Thank you for submitting your manuscript entitled "Linking medicinal cannabis to autotaxin-lysophosphatic acid signalling." to Life Science Alliance. The manuscript was assessed by expert reviewers, whose comments are appended to this letter. We invite you to submit a revised manuscript addressing the Reviewer comments.

Thank you for this interesting contribution to Life Science Alliance. We are looking forward to receiving your revised manuscript.

Sincerely,

B. MANUSCRIPT ORGANIZATION AND FORMATTING:

Reviewer #1 (Comments to the Authors (Required)):

The basis of this paper is to establish a functional interaction of THC with autotaxin-lysophosphatidic acid signaling and to highlight novel aspects of medicinal cannabis therapy. The results and conclusions are potentially interesting but need more justification.

- 1) The kinetic analyses that are used in this paper are not valid for the interpretation of the enzyme kinetics that uses lipid substrates and hydrophobic inhibitors (LPC and the cannabinoids). It has been known for decades in the field of lipid enzymology that conventional kinetics analysis cannot be used because the investigator does not know the relative concentrations of different physical forms of the lipid substrates/inhibitors in the enzyme assay (monomolecular, protein-bound, micellar, liposomes or aggregates etc) and which is the form that interacts with the enzyme. For background see Carman GM, Deems RA, Dennis EA. Lipid signaling enzymes and surface dilution kinetics. J Biol Chem. 1995;270(32):18711-4. All that the authors know is the bulk concentration of LPC and cannabinoids and these bulk values cannot be just entered into the kinetic analysis. The authors should acknowledge these limitations. The EC50 values should be described as "apparent EC50 values".
- 2) The details of the ATX assay are not provided in sufficient detail to repeat the method. Is it correct that albumin was not added to the assay? How was the LPC solution/suspension prepared? How were the solutions/suspensions of the cannabinoids prepared for adding to the assays and what was the basis of using different concentrations of stock solutions for the cannabinoids? What was the vehicle used for the control incubation? These are important considerations that need to be addressed.
- 3) Was a choline standard curve used to standardize the assay? What was the rate of ATX activity (nmol/min/mg ATX) in Fig. 1B and C? This information should be given
- 4) Line 98. Presumably in the presence of choline, which should be mentioned.
- 5) Statistically significant differences should be shown on Fig. 1 B and C and the method of statistical analysis should be described.
- 6) Details of kinetic analysis should be described to demonstrate how the variance of the apparent EC50 values were calculated.
- 7) In interpreting the inhibition curves in Fig. 2, what are the critical micellar concentrations for THC and 6a10a-THC? The curves do not necessarily establish a partial inhibition as also claimed in the Results and Discussion Sections. The failure to inhibit ATX activity completely could reflect the generation of a different physical form of the inhibitor e.g. a micelle, which does not interact with ATX.
- 8) Fig. 2E. It was concluded that the mode of inhibition of THC with ATX- indicates a mixed type inhibition. This type of kinetic interpretation cannot be made with lipid inhibitors that are not defined in terms of physical composition.
- 9) The medium used for measuring LPA1 internalization was described as fatty acid free DMEM to avoid hLPA1 activation by serum. Do the authors mean that the DMEM was fatty acid free (which seems to be strange) or do they mean that fatty acid free albumin was used and if so at what concentration?
- 10) Fig. 5 and Fig. EV5..The concentrations of the reagents used needs to be on the figures or in the legends. More controls needed e.g. a low concentration of LPA plus TLC.
- 11) The authors should strengthen their evidence for the binding of THC to ATX by showing why the range of compounds that do not inhibit should not bind.
- 12) More evidence is required to demonstrate an effect of cannabinoids on ATX activity in vivo because this effect is critical to establish the value of this paper. For example, THC in this saliva has been detected at up to 4167 ng/ml (13 μ M). LPA is present in saliva and ATX expression can be detected in salivary gland tissue. This would provide an ideal system for determining the effects of THC on salivary LPA concentrations and ATX activity.

Minor comment

Line 38 serum should probably be plasma because several other mechanisms of LPA generation are activated during blood clotting.

Reviewer #2 (Comments to the Authors (Required)):

The manuscript by Eymery et al. examines whether Δ^9 -trans-tetrahydrocannabinol (THC), the main psychoactive compound of medicinal cannabis, and derivatives, influences the activity of autotaxin (ATX), a lysophospholipase D. Autotaxin produces lysophosphatidic acid (LPA), which in turn binds to LPA receptors and influences many physiological and pathophysiological processes. Typically, increased ATX-LPA signaling is implicated in disease states including cancer pathogenesis, neuropathic pain, systemic sclerosis, and cardiovascular and metabolic diseases. Thus, identification of well-tolerated inhibitors of ATX could potentially lead to new and improved treatments of these diseases.

In this study, THC and its derivative, 6a10aTHC, were identified as quite potent inhibitors of ATX. Binding interface of these compounds to ATX was examined using X-ray crystallography. In cultured cells, it was shown that these compounds indeed reduce LPA receptor 1 internalization upon ATX and LPC (precursor of LPA) stimulation, indicating impaired ATX activity, and LPA receptor activation and signaling.

These data uncover a new role for THC and 6a10aTHC and provide insight into the potential effect of medicinal cannabis consumption on the ATX-LPA signaling pathway that could be used for therapeutic benefit.

This study is interesting, well-designed, and well-written, and contributes to our understanding of how ATX is regulated by THC and derivatives. Few points for improvement are stated below:

Major Points:

1. Although the inhibitory activity of THC and 6a10aTHC is elegantly shown using purified ATX isoforms and indirect measurement of LPA1 activation in cells (expressing exogenous LPA1), it would perhaps be helpful to understand whether these compounds can lower ATX activity to a similar extent in plasma (from humans or rodent models) where the enzyme is in a more "native" environment reflective of what happens with the ATX-LPA signaling pathway upon medicinal cannabis ingestion in vivo. Alternatively, these compounds could be administered to rodent models followed by the assessment of ATX inhibitory activity.

Minor Points:

1. Sentence in line 121 needs rewording.
2. Line 166: is the "second" isoform of ATX from *Rattus norvegicus* ATX-beta? Please clarify.
3. Line 209: change "in an cellular content" to "in a cellular content".

We thank the reviewers for their time to carefully review our submitted manuscript. We have now completed additional experiments to address most of your major concerns. We have included these new results with all minor corrections suggested in our resubmitted manuscript. These have been highlighted for clarity in the revised manuscript. We believe that these results have significantly improved the quality of our revised manuscript and again thank the reviewers for their comments. Below you can find in more detail our point-by-point response to your comments.

Reviewer #1 (Comments to the Authors (Required)):

The basis of this paper is to establish a functional interaction of THC with autotaxin-lysophosphatidic acid signaling and to highlight novel aspects of medicinal cannabis therapy. The results and conclusions are potentially interesting but need more justification.

1) The kinetic analyses that are used in this paper are not valid for the interpretation of the enzyme kinetics that uses lipid substrates and hydrophobic inhibitors (LPC and the cannabinoids). It has been known for decades in the field of lipid enzymology that conventional kinetics analysis cannot be used because the investigator does not know the relative concentrations of different physical forms of the lipid substrates/inhibitors in the enzyme assay (monomolecular, protein-bound, micellar, liposomes or aggregates etc) and which is the form that interacts with the enzyme. For background see Carman GM, Deems RA, Dennis EA. Lipid signaling enzymes and surface dilution kinetics. *J Biol Chem.* 1995;270(32):18711-4. All that the authors know is the bulk concentration of LPC and cannabinoids and these bulk values cannot be just entered into the kinetic analysis. The authors should acknowledge these limitations. The EC50 values should be described as "apparent EC50 values".

The authors are very grateful for the reviewer's comment on the enzyme kinetics using lipid substrates. The authors acknowledge the problems arising from such kinetics and the EC 50 values have been renamed as "apparent EC50 values" throughout the manuscript.

2) The details of the ATX assay are not provided in sufficient detail to repeat the method. Is it correct that albumin was not added to the assay? How was the LPC solution/suspension prepared? How were the solutions/suspensions of the cannabinoids prepared for adding to the assays and what was the basis of using different concentrations of stock solutions for the cannabinoids? What was the vehicle used for the control incubation? These are important considerations that need to be addressed.

The authors changed the manuscript accordingly and apologize for any unclarity. The paragraph reads now as follows: The LPC solution was obtained by evaporating a commercial LPC chloroform solution directly in a 1.5 mL Eppendorf under vacuum. The dried LPC was then resuspended in water to obtain the mentioned concentration and incubated at 37 °C on a shaker for 5 to 10 min before addition in the microplate. The cannabinoids solutions were prepared from a stock solution in ethanol or other organic solvents. After initial evaporation and/or dilution to obtain the highest concentration mentioned in the assay figure, a 2-fold dilution was performed in Eppendorf tubes. All the compounds were dissolved in 100% ethanol as a vehicle.

*It is correct that no albumin was added to the assays. To address this point we determined the enzymatic activities of ATX in the presence or absence of fatty acid free BSA, which are similar. The authors added a graphical comparison of enzymatic activity in presence or absence of 0.5 mg/mL fatty acid free BSA (FigS6 E and F) for the timescale used during the analysis. Moreover, a recent academic and industrial publication did not use BSA in their assays (Keune WJ et al. *Nat Commun.* 2016 Apr 14;7:11248. doi: 10.1038/ncomms11248. PMID: 27075612; PMCID: PMC4834639) (Hunziker et al., *Front Pharmacol.* 2022 Jan 18;12:699535.).*

3) Was a choline standard curve used to standardize the assay? What was the rate of ATX activity (nmol/min/mg ATX) in Fig. 1B and C? This information should be given

A choline standard curve has now been added to the supplementary information fig S1. An activity rate of 99179,2877 and 65887,0065 μM end product/mM ATX/min has been found for figure 1B and C, respectively, which is similar to the ATX rate previously determined (Saunders et al., Journal of Biological Chemistry. 2011 Aug;286(34):30130–41.).

4) Line 98. Presumably in the presence of choline, which should be mentioned.

Indeed, the control assay was done in presence of choline. This has been corrected in the manuscript and we apologize for the mistake.

5) Statistically significant differences should be shown on Fig. 1 B and C and the method of statistical analysis should be described.

ANOVA comparison between CTRL and other conditions showed statically significant differences for THC, 6a10aTHC, THCA, CBD and 5-DMH-CBD for ATX- β and ATX- γ inhibition (p -value<0.005). ATX was not significantly inhibited by JWH-018, AEA and 2-AG (p -value>0.005). This has now been added in the manuscript.

6) Details of kinetic analysis should be described to demonstrate how the variance of the apparent EC50 values were calculated.

The apparent EC50 determination has been performed with graphpad as follows, and the manuscript has been changed accordingly.

The data analysis was performed with GraphPad (9.4.1). For apparent IC50 determination, fluorescent time points are subtracted from the baseline. From the subtracted results, a linear regression analysis was run on the linear part of the fluorescent curve, between 10 min and 25 min. The linear regression slopes were then plotted and normalized for each inhibitor concentration. A non-linear regression using the following equation was performed with graphpad in order to calculate the IC50:

$$Y = \text{Bottom} + (\text{Top} - \text{Bottom}) / (1 + (\text{IC50}/X)^{\text{HillSlope}})$$

The apparent EC50 was calculated as the concentration of inhibitor that gives a response halfway between maximal and minimal ATX activity. The SEM of the apparent EC50 was determined by graphpad as the 95% confidence interval of the mean.

7) In interpreting the inhibition curves in Fig. 2, what are the critical micellar concentrations for THC and 6a10a-THC? The curves do not necessarily establish a partial inhibition as also claimed in the Results and Discussion Sections. The failure to inhibit ATX activity completely could reflect the generation of a different physical form of the inhibitor e.g. a micelle, which does not interaction with ATX.

To the best of the authors' knowledge a CMC for THC and 6a10aTHC has not been documented in literature.

8) Fig. 2E. It was concluded that the mode of inhibition of THC with ATX- γ indicates a mixed type inhibition. This type of kinetic interpretation cannot be made with lipid inhibitors that are not defined in terms of physical composition.

The author acknowledges this issue. However, it is the most widely used experimental method to determine a mode of inhibition that matches with the structural analysis of ATX co-structures. We also based our work on the article published by Keune et al using the choline release assay to determine a mode of inhibition Keune WJ et al. Nat Commun. 2016 Apr 14;7:11248. doi: 10.1038/ncomms11248. PMID: 27075612; PMCID: PMC4834639) (J. Med. Chem. 2022, 65, 8, 6338–6351 Publication Date: April 20, 2022 <https://doi.org/10.1021/acs.jmedchem.2c00368>).

9) The medium used for measuring LPA1 internalization was described as fatty acid free DMEM to avoid hLPA1 activation by serum. Do the authors mean that the DMEM was fatty acid free (which seems to be strange) or do they mean that fatty acid free albumin was used and if so at what concentration?

1 % (v/v) fatty acid free FBS (Gibco #A3382101) and not albumin only, was added to DMEM as a carrier for LPC.

10) Fig. 5 and Fig. EV5..The concentrations of the reagents used needs to be on the figures or in the legends. More controls needed e.g. a low concentration of LPA plus TLC.

Concentrations have now been added on the top left corner of each image. Additional control experiments have now been performed, including LPA + THC, as well as increased LPC concentration. These have been added to Fig S5. The LPA + THC control experiment confirms that the inhibition of the internalization is not mediated by a THC-LPA1 receptor mechanism, but by the ATX-LPA axis inhibition via THC.

11) The authors should strengthen their evidence for the binding of THC to ATX by showing why the range of compounds that do not inhibit should not bind.

The authors thank the reviewer for this comment. Most of the cannabinoids bind ATX. However, most are weak inhibitors so we focused our work on the most potent ones identified. This can be explained by the similar structural features between cannabinoids and ATX substrates or THC.

12) More evidence is required to demonstrate an effect of cannabinoids on ATX activity in vivo because this effect is critical to establish the value of this paper. For example, THC in this saliva has been detected at up to 4167 ng/ml (13 μ M). LPA is present in saliva and ATX expression can be detected in salivary gland tissue. This would provide an ideal system for determining the effects of THC on salivary LPA concentrations and ATX activity.

The authors were not able to detect ATX-specific activity using the choline release assay with saliva samples from 3 distinct individuals. We were also unable to detect ATX in saliva in preliminary western blot experiments. It might indeed be interesting to see if ATX can be detected in saliva, but we feel that this is out of scope for our current work.

Minor comment

Line 38 serum should probably be plasma because several other mechanisms of LPA generation are activated during blood clotting.

The authors appreciate this valuable comment of the reviewer. Serum was replaced by plasma in line 38 and in line 39.

Reviewer #2 (Comments to the Authors (Required)):

The manuscript by Eymery et al. examines whether Δ 9-trans-tetrahydrocannabinol (THC), the main psychoactive compound of medicinal cannabis, and derivatives, influences the activity of autotaxin (ATX), a lysophospholipase D. Autotaxin produces lysophosphatidic acid (LPA), which in turn binds to LPA

receptors and influences many physiological and pathophysiological processes. Typically, increased ATX-LPA signaling is implicated in disease states including cancer pathogenesis, neuropathic pain, systemic sclerosis, and cardiovascular and metabolic diseases. Thus, identification of well-tolerated inhibitors of ATX could potentially lead to new and improved treatments of these diseases.

In this study, THC and its derivative, 6a10aTHC, were identified as quite potent inhibitors of ATX. Binding interface of these compounds to ATX was examined using X-ray crystallography. In cultured cells, it was shown that these compounds indeed reduce LPA receptor 1 internalization upon ATX and LPC (precursor of LPA) stimulation, indicating impaired ATX activity, and LPA receptor activation and signaling.

These data uncover a new role for THC and 6a10aTHC and provide insight into the potential effect of medicinal cannabis consumption on the ATX-LPA signaling pathway that could be used for therapeutic benefit.

This study is interesting, well-designed, and well-written, and contributes to our understanding of how ATX is regulated by THC and derivatives. Few points for improvement are stated below:

Major Points:

1. Although the inhibitory activity of THC and 6a10aTHC is elegantly shown using purified ATX isoforms and indirect measurement of LPA1 activation in cells (expressing exogenous LPA1), it would perhaps be helpful to understand whether these compounds can lower ATX activity to a similar extent in plasma (from humans or rodent models) where the enzyme is in a more "native" environment reflective of what happens with the ATX-LPA signaling pathway upon medicinal cannabis ingestion in vivo. Alternatively, these compounds could be administered to rodent models followed by the assessment of ATX inhibitory activity.

The authors thank the reviewer for its useful comment. However, ATX activity couldn't be determined using ex vivo experiments, since French law requires a full screening for infectious diseases and it takes 2 days between taking blood and an effective delivery of the samples. Performing in vivo experiment would be of course interesting, however this is beyond scope of the current manuscript, as the authors would need to apply for new licenses for animal work and/or any other experiments with THC than the one granted for biochemical and crystallographic studies.

Minor Points:

1. Sentence in line 121 needs rewording.

The sentence has been reworded and reads now as follows: "However, this artificial cannabinoid did not influence the catalytic activity of either ATX b and g isoforms (Fig. 1 B and C)."

2. Line 166: is the "second" isoform of ATX from *Rattus norvegicus* ATX-beta? Please clarify.

Yes, the 2nd isoform from rATX is the beta isoform. This has been updated in the manuscript.

3. Line 209: change "in an cellular content" to "in a cellular content".

This has been changed in the manuscript.

November 22, 2022

RE: Life Science Alliance Manuscript #LSA-2022-01595-TR

Dr. Andrew A. McCarthy
European Molecular Biology Laboratory
Grenoble
71 avenue des Martyrs
Grenoble Cedex 9 38042
France

Dear Dr. McCarthy,

Thank you for submitting your revised manuscript entitled "Linking medicinal cannabis to autotaxin-lysophosphatidic acid signalling.". We would be happy to publish your paper in Life Science Alliance pending final revisions necessary to meet our formatting guidelines.

- please address the final Reviewer #1's points
- please add ORCID ID for secondary corresponding author-you should have received instructions on how to do so
- please update the figure callouts for Figure S6; the callouts should be Figure S6A,B, etc. rather than EV figure; please add a callout for Figure S6E and S6F to the main manuscript text

Figure Check:

- Figure S5 needs scale bars

A. FINAL FILES:

B. MANUSCRIPT ORGANIZATION AND FORMATTING:

Sincerely,

Reviewer #1 (Comments to the Authors (Required)):

The reviewer thanks the authors for improving their paper. Attention to the following can also help further.

- 1) The use of "apparent EC50 values" is a great improvement. Perhaps it would help to read to explain why this is done so that others are able to benefit by the explanation.
- 2) Line 143. The units of kinetic activity, which are listed as 99179 and 65887 μM end product/mM ATX/min seems to be unusual because units such as moles are normally used rather than moles/L. Do the authors really mean moles/L and also do they also claim to be able to measure activity to more than three significant figures?

Reviewer #2 (Comments to the Authors (Required)):

The reviewer queries have been addressed appropriately.

November 25, 2022

RE: Life Science Alliance Manuscript #LSA-2022-01595-TRR

Dr. Andrew A. McCarthy
European Molecular Biology Laboratory
Grenoble
71 avenue des Martyrs
Grenoble Cedex 9 38042
France

Dear Dr. McCarthy,

Thank you for submitting your Research Article entitled "Linking medicinal cannabis to autotaxin-lysophosphatidic acid signaling.". It is a pleasure to let you know that your manuscript is now accepted for publication in Life Science Alliance. Congratulations on this interesting work.

DISTRIBUTION OF MATERIALS:

Again, congratulations on a very nice paper. I hope you found the review process to be constructive and are pleased with how the manuscript was handled editorially. We look forward to future exciting submissions from your lab.

Sincerely,
